# Deep Learning for Real-Time Atari Game Play Using Offline Monte-Carlo Tree Search Planning

**Xiaoxiao Guo**
Computer Science and Eng.
University of Michigan
guoxiao@umich.edu

**Satinder Singh**
Computer Science and Eng.
University of Michigan
baveja@umich.edu

**Honglak Lee**
Computer Science and Eng.
University of Michigan
honglak@umich.edu

**Richard Lewis**
Department of Psychology
University of Michigan
rickl@umich.edu

**Xiaoshi Wang**
Computer Science and Eng.
University of Michigan
xiaoshiw@umich.edu

## Abstract

The combination of modern Reinforcement Learning and Deep Learning approaches holds the promise of making significant progress on challenging applications requiring both rich perception and policy-selection. The Arcade Learning Environment (ALE) provides a set of Atari games that represent a useful benchmark set of such applications. A recent breakthrough in combining model-free reinforcement learning with deep learning, called DQN, achieves the best real-time agents thus far. Planning-based approaches achieve far higher scores than the best model-free approaches, but they exploit information that is not available to human players, and they are orders of magnitude slower than needed for real-time play. Our main goal in this work is to build a better real-time Atari game playing agent than DQN. The central idea is to use the slow planning-based agents to provide training data for a deep-learning architecture capable of real-time play. We proposed new agents based on this idea and show that they outperform DQN.

## 1    Introduction

Many real-world Reinforcement Learning (RL) problems combine the challenges of closed-loop action (or policy) selection with the already significant challenges of high-dimensional perception (shared with many Supervised Learning problems). RL has made substantial progress on theory and algorithms for policy selection (the distinguishing problem of RL), but these contributions have not directly addressed problems of perception. Deep learning (DL) approaches have made remarkable progress on the perception problem (e.g., [11, 17]) but do not directly address policy selection. RL and DL methods share the aim of generality, in that they both intend to minimize or eliminate domain-specific engineering, while providing "off-the-shelf" performance that competes with or exceeds systems that exploit control heuristics and hand-coded features. Combining modern RL and DL approaches therefore offers the potential for general methods that address challenging applications requiring both rich perception and policy-selection.

The Arcade Learning Environment (ALE) is a relatively new and widely accessible class of benchmark RL problems that provide a particularly challenging combination of policy selection and perception. ALE includes an emulator and a large number of Atari 2600 (a 1970s–80s home video console) games. The complexity and diversity of the games—both in terms of perceptual challenges in mapping pixels to useful features for control and in terms of the control policies needed—make

ALE a useful set of benchmark RL problems, especially for evaluating general methods intended to achieve success without hand-engineered features.

Since the introduction of ALE, there have been a number of attempts to build general-purpose Atari game playing agents. The departure point for this paper is a recent and significant breakthrough [16] that combines RL and DL to build agents for multiple Atari Games. It achieved the best machine-agent real-time game play to date (in some games close to or better than human-level play), does not require feature engineering, and indeed reuses the same perception architecture and RL algorithm across all the games. We believe that continued progress on the ALE environment that preserves these advantages will extend to broad advances in other domains with significant perception and policy selection challenges. Thus, our immediate goal in the work reported here is to build even better performing general-purpose Atari Game playing agents. We achieve this by introducing new methods for combining RL and DL that use slow, off-line Monte Carlo tree search planning methods to generate training data for a deep-learned classifier capable of state-of-the-art real-time play.

## 2    Brief background on RL and DL and challenges of perception

RL and more broadly decision-theoretic planning has a suite of methods that address the challenge of selecting/learning good policies, including value function approximation, policy search, and Monte-Carlo Tree Search [9, 10] (MCTS). These methods have different strengths and weaknesses and there is increasing understanding of how to match them to different types of RL-environments. Indeed, an accumulating number of applications attest to this success. But it is still not the case that there are reasonably off-the-shelf approaches to solving complex RL problems of interest to Artificial Intelligence (AI) such as the games in ALE. One reason for this is that despite major advances there hasn't been an off-the-shelf approach to significant perception problems. The perception problem itself has two components: 1) the sensors at any time step do not capture all the information in the history of observations, leading to partial observability, and 2) the sensors provide very high-dimensional observations that introduce computational and sample-complexity challenges for policy selection.

One way to handle the perception challenges when a model of the RL environment is available is to avoid the perception problem entirely by eschewing the building of an explicit policy and instead using repeated incremental planning via MCTS methods such as UCT [10] (discussed below). Either when a model is not available, or when an explicit representation of the policy is required, the usual approach to applied RL success has been to use expert-developed task-specific features of a short history of observations in combination with function approximation methods and some trial-and-error on the part of the application developer (on small enough problems this can be augmented with some automated feature selection methods). Eliminating the dependence of applied RL success on engineered features motivates our interest in combining RL and DL (though see [20] for early work in this direction).

Over the past decades, deep learning (see [3, 19] for a survey) has emerged as a powerful technique for *learning* feature representations from data (again, this is in a stark contrast to the conventional way of hand-crafting features by domain experts). For example, DL has achieved state-of-the-art results in image classification [11, 4], speech recognition [15, 17, 6], and activity recognition [12, 8]. In DL, features are learned in a compositional hierarchy. Specifically, low-level features are learned to encode low-level statistical dependencies (e.g., "edges" in images), and higher-level features encode higher-order dependencies of the lower-level features (e.g., "object parts") [14]. In particular, for data that has strong spatial or temporal dependencies, convolutional neural networks [13] have been shown to learn invariant high-level features that are informative for supervised tasks. Such convolutional neural networks were used in the recent successful combination of DL and RL for Atari Game playing [16] that forms the departure point of our work. We describe this work in more detail below.

## 3    Existing Work on Atari Games and a Performance Gap

While the games in ALE are simpler than many modern games, they still pose significant challenges to human players. In RL terms, for a human player these games are Partially-Observable Markov Decision Processes (POMDPs). The true state of each game at any given point is captured by the

contents of the limited random-access memory (RAM). A human player does not observe the state and instead perceives the game screen (frame) which is a 2D array of 7-bit pixels, 160 pixels wide by 210 pixels high. The action space available to the player depends on the game but maximally consists of the 18 discrete actions defined by the joystick controller. The next state is a deterministic function of the previous state and the player's action choice. Stochasticity in these games is limited to the choice of the initial state of the game (which can include a random number seed stored in RAM). So even though the state transitions are deterministic, the transitions from history of observations and actions to next observation can be stochastic (because of the stochastic initial hidden state). The immediate reward at any given step is defined by the game and made available by the ALE; it is usually a function of the current frame or the difference between current and previous frames. When running in real-time, the simulator generates 60 frames per second. All the games we consider terminate in a finite number of time-steps (and so are episodic). The goal in these games is to select an optimal policy, i.e., to select actions in such a way so as to maximize the expected value of the cumulative sum of rewards until termination.

**Model-Free RL Agents for Atari Games.** Here we discuss work that does **not** access the state in the games and thus solves the game as a POMDP. In principle one could learn a state representation and infer an associated MDP model using frame-observation and action trajectories, but these games are so complex that this is rarely done. Instead, partial observability is dealt with by hand-engineering features of short histories of frames observed so far and model-free RL methods are used to learn good policies as a function of those feature representations. For example, the paper that introduced ALE [1], used SARSA with several different hand-engineered features sets. The contingency awareness approach [4] improved performance of the SARSA algorithm by augmenting the feature sets with a learned representation of the parts of the screen that are under the agent's control. The sketch-based approach [2] further improves performance by using the tug-of-war sketch features. HyperNEAT-GGP [7] introduces an evolutionary policy search based Atari game player. Most recently Deep Q-Network (hereafter DQN) [16] uses a modified version of Q-Learning with a convolutional neural network (CNN) with three hidden layers for function approximation. This last approach is the state of the art in this class of methods for Atari games and is the basis for our work; we present the relevant details in Section 5. It does not use hand-engineered features but instead provides the last four raw frames as input (four instead of one to alleviate partial observability).

**Planning Agents for Atari Games based on UCT.** These approaches access the state of the game from the emulator and hence face a deterministic MDP (other than the random choice of initial state). They incrementally plan the action to take in the current state using UCT, an algorithm widely used for games. UCT has three parameters, the number of trajectories, the maximum-depth (uniform for each trajectory), and a exploration parameter (a scalar set to 1 in all our experiments). In general, the larger the trajectory & depth parameters are, the slower UCT is but the better it is. UCT uses the emulator as a model to simulate trajectories as follows. Suppose it is generating the $k^{th}$ trajectory and the current node is at depth $d$ and the current state is $s$. It computes a score for each possible action $a$ in state-depth pair $(s, d)$ as the sum of two terms, an exploitation term that is the Monte-Carlo average of the discounted sum of rewards obtained from experiences with state-depth pair $(s, d)$ in the previous $k - 1$ trajectories, and an exploration term that is $\sqrt{\log{(n(s, d))}/n(s, a, d)}$ where $n(s, a, d)$ and $n(s, d)$ are the number of experiences of action $a$ with state-depth pair $(s, d)$ and with state-depth pair $(s, d)$ respectively in the previous $k - 1$ trajectories. UCT selects the action to simulate in order to extend the trajectory greedily with respect to this summed score. Once the input-parameter number of trajectories are generated each to maximum depth, UCT returns the exploitation term for each action at the root node (which is the current state it is planning an action for) as its estimate of the utility of taking that action in the current state of the game. UCT has the nice theoretical property that the number of simulation steps (number of trajectories × maximum-depth) needed to ensure any bound on the loss of following the UCT-based policy is independent of the size of the state space; this result expresses the fact that the use of UCT avoids the perception problem, but at the cost of requiring substantial computation for every time step of action selection because it never builds an explicit policy.

**Performance Gap & our Opportunity.** The opportunity for this paper arises from the following observations. The model-free RL agents for Atari games are fast (indeed faster than real-time, e.g., the CNN-based approach from our paper takes $10^{-4}$ seconds to select an action on our computer) while the UCT-based planning agents are several orders of magnitude slower (much slower than real-time, e.g., they take seconds to select an action on the same computer). On the other hand,

the performance of UCT-based planning agents is much better than the performance of model-free RL agents (this will be evident in our results below). Our goal is to develop methods that retain the DL advantage of not needing hand crafted features and the online real-time play ability of the model-free RL agents by exploiting data generated by UCT-planning agents.

# 4 Methods for Combining UCT-based RL with DL

We first describe the baseline UCT agent, and then three agents that instantiate different methods of combining the UCT agent with DL. Recall that in keeping with the goal of building general-purpose methods as in the DQN work we impose the constraint of reusing the same input representations, the same function approximation architecture, and the same planning method for all the games.

## 4.1 Baseline UCT agent that provides training data

This agent requires no training. It does, however, require specification of its two parameters, the number of trajectories and the maximum-depth. Recall that our proposed new agents will all use data from this UCT-agent to train a CNN-based policy and so it is reasonable that the resulting performance of our proposed agents will be worse than that of the UCT-agent. Therefore, in our experiments we set these two parameters large enough to ensure that they outscore the published DQN scores, but not so large that they make our computational experiments unreasonably slow. Specifically, we elected to use $300$ as maximum-depth and $10000$ as number of trajectories for all games but two. Pong turns out to be a much simpler game and we could reduce the number of trajectories to $500$, and Enduro turned out to have more distal rewards than the other games and so we used a maximum-depth of $400$. As will be evident from the results in Section 5 this allowed the UCT agent to significantly outperform DQN in all games but Pong in which DQN already performs perfectly. We emphasize that the UCT agent does not meet our goal of real-time play. For example, to play a game just $800$ times with the UCT agent (we do this to collect training data for our agent's below) takes a few days on a recent multicore computer for each game.

## 4.2 Our three methods and their corresponding agents

**Method 1: UCTtoRegression** (for **UCT** to **C**NN via **R**egression). The key idea is to use the action values computed by the UCT-agent to train a regression-based CNN. The following is done for each game. Collect $800$ UCT-agent runs by playing the game $800$ times from start to finish using the UCT agent above. Build a dataset (table) from these runs as follows. Map the last four frames of each state along each trajectory into the action-values of all the actions as computed by UCT. This training data is used to train the CNN via regression (see below for CNN details). The UCTtoRegression-agent uses the CNN learned by this training procedure to select actions during evaluation.

**Method 2: UCTtoClassification** (for **UCT** to **C**NN via **C**lassification). The key idea is to use the action choice computed by the UCT-agent (selected greedily from action-values) to train a classifier-based CNN. The following is done for each game. Collect $800$ UCT-agent runs as above. These runs yield a table in which the rows correspond to the last four frames at each state along each trajectory and the single column is the choice of action that is best according to the UCT-agent at that state of the trajectory. This training data is used to train the CNN via multinomial classification (see below for CNN details). The UCTtoClassification-agent uses the CNN-classifier learned by this training procedure to select actions during evaluation.

One potential issue with the above two agents is that the training data's input distribution is generated by the UCT-agent while during testing the UCTtoRegression and UCTtoClassification agents will perform differently from the UCT-agent and thus could experience an input distribution quite difference from that of the UCT-agent's. This could limit the testing performance of the UCTtoRegression and UCTtoClassification agents. Thus, it might be desirable to somehow bias the distribution over inputs to those likely to be encountered by these agents; this observation motivates our next method.

**Method 3: UCTtoClassification-Interleaved** (for **UCT** to **C**NN via **C**lassification-**I**nterleaved). The key idea is to focus UCT planning on that part of the state space experienced by the (partially trained) CNN player. The method accomplishes this by interleaving training and data collection as

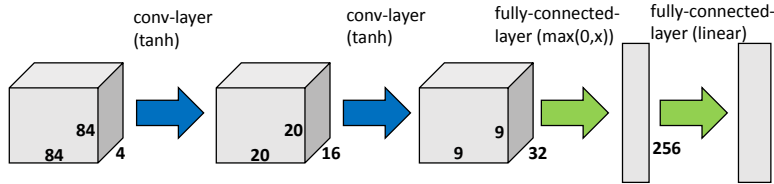

Figure 1: The CNN architecture from DQN [6] that we adopt in our agents. See text for details.

follows[1]. Collect 200 UCT-agent runs as above; these will obviously have the same input distribution concern raised above. The data from these runs is used to train the CNN via multinomial classification just as in the UCTtoClassification-agent's method (we do not do this for the UCTtoRegression-agent because as we show below it performs worse than the UCTtoClassification-agent). The trained CNN is then used to decide action choices in collecting a further 200 runs (though 5% of the time a random action is chosen to ensure some exploration). At each state of the game along each trajectory, UCT is asked to compute its choice of action and the original data set is augmented with the last four frames for each state as the rows and the column as UCT's action choice. This 400 trajectory dataset's input distribution is now potentially different from that of the UCT-agent. This dataset is used to train the CNN again via multinomial classification. This interleaved procedure is repeated until there are a total of 800 runs worth of data in the dataset for the final round of training of the CNN. The UCTtoClassification-Interleaved agent uses the final CNN-classifier learned by this training procedure to select actions during testing.

In order to focus our empirical evaluation on the contribution of the non-DL part of our three new agents, we reused exactly the same convolutional neural network architecture as used in the DQN work (we describe this architecture in brief detail below). The DQN work modified the reward functions for some of the games (by saturating them at $+1$ and $-1$) while we use unmodified reward functions (these only play a role in the UCT-agent components of our methods and not in the CNN component). We also follow DQN's frame-skipping techniques: the agent sees and selects actions on every $k^{th}$ frame instead of every frame ($k = 3$ for Space Invaders and $k = 4$ for all other games), and the latest chosen-action is repeated on subsequently-skipped frames.

### 4.3 Details of Data Preprocessing and CNN Architecture

**Preprocessing** (identical to DQN to the best of our understanding). Raw Atari game frames are $160 \times 210$ pixel images with a 128-color palette. We convert the RGB representation to gray-scale and crop an $160 \times 160$ region of the image that captures the playing area, and then the cropped image is down-sampled to $84 \times 84$ in order to reuse DQN's CNN architecture. This procedure is applied to the last 4 frames associated with a state and stacked to produce a $84 \times 84 \times 4$ preprocessed input representation for each state. We subtracted the pixel-level means and scale the inputs to lie in the range [-1, 1]. We shuffle the training data to break the strong correlations between consecutive samples, which therefore reduces the variance of the updates.

**CNN Architecture.** We use the same deep neural network architecture as DQN [16] for our agents. As depicted in Figure 1, our network consists of three hidden layers. The input to the neural network is an $84 \times 84 \times 4$ image produced by the preprocessing procedure above. The first hidden layer convolves 16, $8 \times 8$, filters with stride 4 with the input image and applies a rectifier nonlinearity (tanh). The second hidden layer convolves 32, $4 \times 4$, filters with stride 2 again followed by a rectifier nonlinearity (tanh). The final hidden layer is fully connected and consists of 256 rectifier (max) units. In the multi-regression-based agent (UCTtoRegression), the output layer is a fully connected linear layer with a single output for each valid action. In the classification-based agents (UCTtoClassification, UCTtoClassification-Interleaved), a softmax (instead of linear) function is applied to the final output layer. We refer the reader to the DQN paper for further detail.

Table 1: Performance (game scores) of the four real-time game playing agents, where UCR is short for UCT-toRegression, UCC is short for UCTtoClassification, and UCC-I is short for UCTtoClassification-Interleaved.

| Agent | B.Rider | Breakout | Enduro | Pong | Q*bert | Seaquest | S.Invaders |
|---|---|---|---|---|---|---|---|
| **DQN** | 4092 | 168 | 470 | 20 | 1952 | 1705 | 581 |
| *-best* | 5184 | 225 | 661 | 21 | 4500 | 1740 | 1075 |
| **UCC** | 5342 (20) | 175(5.63) | 558(14) | 19(0.3) | 11574(44) | 2273(23) | 672(5.3) |
| *-best* | 10514 | 351 | 942 | 21 | 29725 | 5100 | 1200 |
| *-greedy* | 5676 | 269 | 692 | 21 | 19890 | 2760 | 680 |
| **UCC-I** | 5388(4.6) | 215(6.69) | 601(11) | 19(0.14) | 13189(35.3) | 2701(6.09) | 670(4.24) |
| *-best* | 10732 | 413 | 1026 | 21 | 29900 | 6100 | 910 |
| *-greedy* | 5702 | 380 | 741 | 21 | 20025 | 2995 | 692 |
| **UCR** | 2405(12) | 143(6.7) | 566(10.2) | 19(0.3) | 12755(40.7) | 1024 (13.8) | 441(8.1) |

Table 2: Performance (game scores) of the off-line UCT game playing agent.

| Agent | B.Rider | Breakout | Enduro | Pong | Q*bert | Seaquest | S.Invaders |
|---|---|---|---|---|---|---|---|
| **UCT** | 7233 | 406 | 788 | 21 | 18850 | 3257 | 2354 |

# 5 Experimental Results

First we present our main performance results and then present some visualizations to help understand the performance of our agents. In Table 1 we compare and contrast the performance of the four real-time game playing agents, three of which (UCTtoRegression, UCTtoClassification, and UCTtoClassification-Interleaved) we implemented and evaluated; the performance of the DQN was obtained from [16].

The columns correspond to the seven games named in the header, and the rows correspond to different assessments of the four agents. Throughout the numbers in parentheses are standard-errors. The DQN row reports the average performance (game score) of the DQN agent (a random action is chosen 5% of the time during testing). The DQN-*best* row is the best performance of the DQN over all the attempts at each game incorporated in the row corresponding to DQN. Comparing the performance of the UCTtoClassification and UCTtoRegression agents (both use 5% exploration), we see that the UCTtoClassification agent either competes well with or significantly outperforms the UCT-toRegression agent. More importantly the UCTtoClassification agent outperforms the DQN agent in all games but Pong (in which both agents do nearly perfectly because the maximum score in this game is 21). In some games (B.Rider, Enduro, Q*Bert, Sequest and S.Invaders) the percentage-performance gain of UCTtoClassification over DQN is quite large. Similar gains are obtained in the comparison of UCTtoClassification-*best* to DQN-*best*.

We used 5% exploration in our agents to match what the DQN agent does, but it is not clear why one should consider random action selection during testing. In any case, the effect of this randomness in action-selection will differ across games (based, e.g., on whether a wrong action can be terminal). Thus, we also present results for the UCTtoClassification-*greedy* agent in which we don't do any exploration. As seen by comparing the rows corresponding to UCTtoClassification and UCTtoClassification-*greedy*, the latter agent always outperforms the former and in four games (Breakout, Enduro, Q*Bert, and Seaquest) achieves further large-percentage improvements.

Table 2 gives the performance of our non-realtime UCT agent (again, with 5% exploration). As discussed above we selected UCT-agent's parameters to ensure that this agent outperforms the DQN agent allowing room for our agents to perform in the middle.

Finally, recall that the UCTtoClassification-Interleaved agent was designed so that its input distribution during training is more likely to match its input distribution during evaluation and we hypothesized that this would improve performance relative to UCTtoClassification. Indeed, in all games but B. Rider, Pong and S.Invaders in which the two agents perform similarly, UCTtoClassification-Interleaved significantly outperforms UCTtoClassification. The same holds when comparing

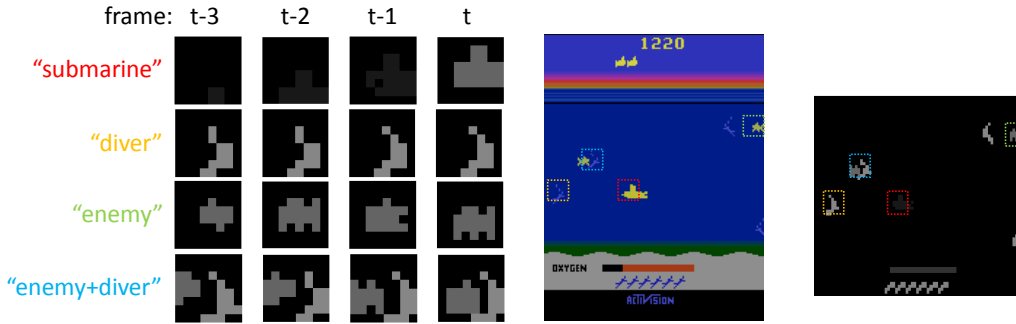

Figure 2: Visualization of the first-layer features learned from Seaquest. (Left) visualization of four first-layer filters; each filter covers four frames, showing the spatio-temporal template. (Middle) a captured screen. (Right) gray-scale version of the input screen which is fed into the CNN. Four filters were color-coded and visualized as dotted bounding boxes at the locations where they get activated. This figure is best viewed in color.

UCTtoClassification-Interleaved-best and UCTtoClassification-best as well as UCTtoClassification-Interleaved-greedy and UCTtoClassification-greedy.

In a further preliminary exploration of the effectiveness of the UCTtoClassification-Interleaved in exploiting additional computational resources for generating UCT runs, on the game Enduro we compared UCTtoClassification and UCTtoClassification-Interleaved where we allowed each of them twice the number of UCT runs used in producing the Table 1 above, i.e., 1600 runs while keeping a batch size of 200. The performance of UCTtoClassification improves from 558 to 581 while the performance of UCTtoClassification-Interleaved improves from 601 to 670, i.e., the interleaved method improved more in absolute and percentage terms as we increased the amount of training data. This is encouraging and is further confirmation of the hypothesis that motivated the interleaved method, because the interleaved input distribution would be even more like that of the final agent with the larger data set.

**Learned Features from Convolutional Layers.** We provide visualizations of the learned filters in order to gain insights on what the CNN learns. Specifically, we apply the "optimal stimuli" method [5] to visualize the features CNN learned after training. The method picks the input image patches that generate the greatest responsive after convolution with the trained filters. We select 8*8*4 input patches to visualize the first convolutional layer features and 20*20*4 to visualize the second convolutional layer filters. Note that these patch sizes correspond to receptive field sizes of the learned features in each layer.

In Figure 2, we show four first-layer filters of the CNN trained from Seaquest for UCTtoClassification-agent. Specifically, each filter covers four frames of 8*8 pixels, which can be viewed as a spatio-temporal template that captures specific patterns and their temporal changes. We also show an example screen capture and

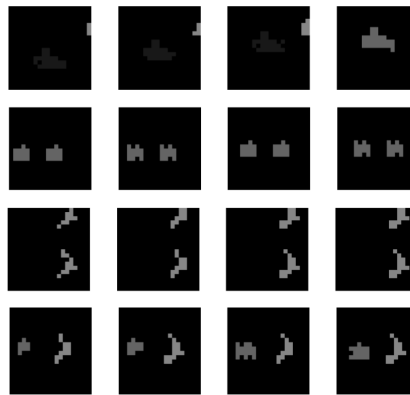

Figure 3: Visualization of the second-layer features learned from Seaquest.

visualize where the filters get activated in the gray-scale version of the image (which is the actual input to the CNN model). The visualization suggests that the first-layer filters capture "object-part" patterns and their temporal movements.

Figure 3 visualizes the four second-layer features via the optimal stimulus method, where each row corresponds to a filter. We can see that the second-layer features capture bigger spatial patterns (often covering beyond the size of individual objects), while encoding interactions between objects, such as two enemies moving together, and submarine moving along a direction. Overall, these qualitative results suggest that the CNN learns relevant patterns useful for game playing.

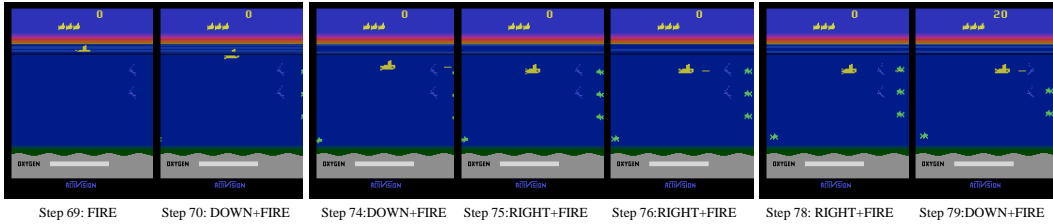

| Step 69: FIRE | Step 70: DOWN+FIRE | Step 74:DOWN+FIRE | Step 75:RIGHT+FIRE | Step 76:RIGHT+FIRE | Step 78: RIGHT+FIRE | Step 79:DOWN+FIRE |

Figure 4: A visualization of the UCTtoClassification agent's policy as it kills an enemy agent.

**Visualization of Learned Policy.** Here we present visualizations of the policy learned by the UCT-toClassification agent with the aim of illustrating both what it does well and what it does not.

Figure 4 shows the policy learned by UCTtoClassification to destroy nearby enemies. The CNN changes the action from "Fire" to "Down+Fire" at time step 70 when the enemies first show up at the right columns of the screen, which will move the submarine to the same horizontal position of the closest enemy. At time step 75, the submarine is at the horizontal position of the closest enemy and the action changes to "Right+Fire". The "Right+Fire" action is repeated until the enemy is destroyed at time step 79. At time step 79, the predicted action is changed to "Down+Fire" again to move the submarine to the horizontal position of the next closest enemy. This shows the UCTtoClassification agent's ability to deal with delayed reward as it learns to take a sequence of unrewarded actions before it obtains any reward when it finally destroys an enemy.

Figure 4 also shows a shortcoming in the UCTtoClassification agent's policy, namely it does not purposefully take actions to save a diver (saving a diver can lead to a large reward). For example, at time step 69, even though there are two divers below and to the right of the submarine (our agent), the learned policy does not move the submarine downward. This phenomenon was observed frequently. The reason for this shortcoming is that it can take a large number of time steps to capture 6 divers and bring them to surface (bringing fewer divers to the surface does not yield a reward); this takes longer than the planning depth of UCT. Thus, it is UCT that does not purposefully save divers and thus the training data collected via UCT reflects that defect which is then also present in the play of the UCTtoClassification (and UCTtoClassification-Interleaved) agent.

## 6   Conclusion

UCT-based planning agents are unrealistic for Atari game play in at least two ways. First, to play the game they require access to the state of the game which is unavailable to human players, and second they are orders of magnitude slower than realtime. On the other hand, by slowing the game down enough to allow UCT to play leads to the highest scores on the games they have been tried on. Indeed, by allowing UCT more and more time (and thus allowing for larger number of trajectories and larger maximum-depth) between moves one can presumably raise the score more and more. We identified a gap between the UCT-based planning agents performance and the best realtime player DQN's performance and developed new agents to partially fill this gap. Our main applied result is that at the time of the writing of this paper we have the best realtime Atari game playing agents on the same 7 games that were used to evaluate DQN. Indeed, in most of the 7 games our best agent beats DQN significantly. Another result is that at least in our experiments training the CNN to learn a classifier that maps game observations to actions was better than training the CNN to learn a regression function that maps game observations to action-values (we intend to do further work to confirm how general this result is on ALE). Finally, we hypothesized that the difference in input distribution between the UCT agent that generates the training data and the input distribution experienced by our learned agents would diminish performance. The UCTtoClassification-Interleaved agent we developed to deal with this issue indeed performed better than the UCTtoClassification agent indirectly confirming our hypothesis and solving the underlying issue.

**Acknowledgments.** This work was supported in part by NSF grant IIS-1148668. Any opinions, findings, conclusions, or recommendations expressed here are those of the authors and do not necessarily reflect the views of the sponsors.

## Footnotes

[1]Our UCTtoClassification-Interleaved method is a special case of DAgger [18] (in the use of a CNN-classifier and in the use of specific choices of parameters $\beta_1 = 1$, and for $i > 1$, $\beta_i = 0$). As a small point of difference, we note that our emphasis in this paper was in the use of CNNs to avoid the use of hand-crafted domain specific features, while the empirical work for DAgger did not have the same emphasis and so used handcrafted features.

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
