[Reviews · NeurIPS 2014]

Submitted by Assigned_Reviewer_17

Review of submission 1706:
Deep Learning for Real-Time Atari Game Play Using Offline Monte-Carlo Tree Search Planning

Summary:

A fast, deep NN is trained to play ALE Atari games, where the teacher is a high-quality, but slow, traditional game planner. This works better than a recent method [19] (here called "DQN") using temporal difference-based reinforcement learning for a deep NN function approximator with the same architecture.

Comments:

Interesting work. I like the simplicity of the basic approach. It is good that somebody implemented this.

Abstract and text are rather verbose though - I suggest to greatly shorten this, maybe in the style of the brief summary above.

Text on previous work: "Over the last decade, deep learning (e.g., [13, 12, 18, 8]; see [7] for a survey) has emerged as a powerful technique for learning feature representations from data (again, this is in a stark contrast to the conventional way of hand-crafting features by domain experts)."

The text above is rather misleading, and its references are rather unbalanced - they almost exclusively refer to recent papers from a few related groups, without pointing out that successful deep learning goes back much further. For example, the cited "survey [7]" focuses on certain results since 2006. However, deep learning of feature representations in neural networks (and similar systems) is much older - a recent comprehensive survey http://arxiv.org/abs/1404.7828 summarizes deep learning methods since 1965.

General recommendation:

The submission is interesting although it confirms what one might expect: slow but good traditional game players can train much faster deep networks in supervised fashion to outperform similar deep networks trained by more general reinforcement learning methods. Publishable, provided a more balanced account of deep learning history is provided.

Summary: The submission is interesting although it confirms what one might expect: slow but good traditional game players can train much faster deep networks in supervised fashion to outperform similar deep networks trained by more general reinforcement learning methods. Publishable, provided a more balanced account of deep learning history is provided.

Submitted by Assigned_Reviewer_37

Paper proposes to use neural networks (generic function approximators)
to learn mapping coming from UCT. UCT is a Monte Carlo method, which
considers multiple trajectories of game. Monte Carrlo methods are
expensive due to consideration of exponential number of trajectories.
Neural networks trained on such mapping allow to retrive results
within orders of magnitude faster time.

Paper is difficult to understand due to many abbreviations. However,
overall concept is quite simple. It seems that this technique could be
used to speed up any Monte Carlo players, but paper describes only
results for UCT. It would be nice, if you would show for other Monte
Carlo methods that this approach works.

Please add some mathematical equations to make paper more clear.

BTW: Would it be possible to train a single model to play all the
games ? Would it help across games ?
Summary: Results are compelling, however presentation can be improved. Authors
should include a some mathematical formulas, and fix figures (e.g.
figure number 4 is quite to read, and understand).

Submitted by Assigned_Reviewer_42

Summary.

This paper studies a number of variations on the topic of training a deep network using data generated by a Monte-Carlo Tree Search (MCTS) agent. The paper focuses on the Atari 2600 platform and is motivated by the observation that, while MCTS performs extremely well on Atari 2600 games, it is also too computationally expensive to be used in a realistic setting. The authors provide empirical results on a number of Atari 2600 games.

Overall.

To me, the main contribution of this paper is to propose to ``compile'' the UCT value function (or policy, according to the algorithm used) into a deep network. The paper is clear, and the results of good quality, but the work lacks in significance. While there are some nice results -- and performance improvements on all but one game -- I feel the topic is simply not sufficiently explored. There is little insight gained into how these results might carry over to other domains, or significant algorithmic improvements that could result from this work.

A few major hurdles:
. The UCC-I algorithm, described here, is very reminiscent of the DAgger algorithm of Ross et al. (2011). It would probably be good to discuss the relationship. Do we expect better sampling guarantees than their approach?
. Are UCR/UCC really valid competitors to previous learning approaches? They still require full access to the simulator.
. The fact that the empirical results rely on game-specific tunings somewhat devalues said results. Are the results so different when a uniform environment parametrization is used?

There are a number of research directions which I believe could improve the paper:
. Further studying how well a policy/value function can be summarized into a deep network. Which is easier? Your results hint that it is better to summarize the policy. Does this tell us something about the nature of Atari 2600 environments?
. What about performing regression on the empirical return, rather than the UCT value? Is there a sense in which one is better than the other?
. The UCT agent doesn't use "features", but these might emerge from the "policy compilation". A related idea was studied by Cobo et al. (2013) in learning features from expert data. Can something like this be investigated here as well?
. How is partial observability handled? The optimal stimuli plots hint at some of this, but it still seems an understudied question.

Minor points:
. line 120: "two broad classes of approaches": this seems to suggest no other way would be valid. Can this be rephrased?
. line 136-143: it would be nice if the related work section discussed how the non-deep network features differ from the new stuff. For example: is the main distinction that the visual features are learned, or that we are using a deep architecture?
. line 136: I believe cite [4] is incorrect in the bibliography. Other cites also seemed incorrectly numbered.
Summary: Overall an interesting idea. What I find missing most from this paper is a main thesis -- a cogent theory to be investigated. A number of decent results are provided, but these would need to be studied more thoroughly to warrant acceptance.
Author Feedback
Author rebuttal: We thank the reviewers for many suggestions and comments and for appreciating the potential impact of our work. We will revise the paper by fixing the typos/phrasing-issues/citation-issues pointed to by the reviewers.

In addition, we will make our code available for the research community.

R17, R37:
We will tighten the abstract & other text to make room for A) a more comprehensive and balanced review of deep-learning work, particularly older foundational work (e.g., as described in Schmidhuber, 2014), B) make better acronyms for improved readability, C) add mathematical detail where helpful.

R42:
On game-specific tuning, the two sets of parameters that varied across games were: 1) The number of frames skipped. We used exactly our competitors (DQNs) choice in order to make a fair comparison. They skipped 4 frames on all games but one in which because of the blinking nature of key objects an even-skip-number could not be used and so they and we skipped 3 frames, and 2) UCT hyper parameters --- our constraint was that these had to be high enough to lead UCT to outperform DQN but low enough that data-collection was computationally feasible (because UCT is so slow). We used the same hyperparameters (1000 trajectories of 300 depth) in all but 2 games. In Pong we used 500 trajectories of 300 depth because those settings were faster and UCT was already optimal with these parameters. In Enduro we used 1000 trajectories of depth 400 because the extra depth was needed to outperform DQN. Thus, we respectfully argue that this is very little game-specific tuning, and if we set the hyperparameters to the same large values (e.g., 1000 trajectories of 400 depth), we expect the performance would remain the same or even slightly improve. We will make this more clear in a revision.

On UCC/UCR being valid competitors. In testing phase (i.e., in play after training) they don’t need access to the simulator at all and so they meet our goal of building a similarly-real-time player as DQN. In training phase, the UCT component does need the simulator to generate the training data. (A subtle aspect is that the simulator is only used to allow a reset so that multiple actions can be tried in a state in building out the UCT trajectories; the contents of RAM are not used in any other way by UCT).

Connection between UCC-I and DAgger. Many thanks for the pointer! DAgger was developed to make a nice theoretical claim. Indeed, UCC-I is a special case of DAgger (in the use of a CNN-classifier and in the use of a specific choice of parameters \beta_{1}=1, and for i > 1, \beta_{i} = 0). Our focus here was on improving the empirical performance of our player by incrementally adapting the distribution over states and it worked well (UCC-I outperformed UCC). Since the submission, we have additional UCC-I versus UCC results showing that doubling the size of the training data does not help UCC much (because the data is of the wrong distribution) but because of the interleaved dataset generation UCC-I improves significantly by doubling the size of the training set. We will add these additional results for UCC-I and cite and discuss the clear connection to DAgger in a revision (as a very small point of difference, we note our emphasis in this paper was in the use of CNNs to avoid the use of handcrafted domain specific features, while the empirical work for DAgger did not have the same emphasis and so used handcrafted features).

On handling partial observability. Exactly as in the DQN work, again to make a tight comparison and allow credit-assignment to the simple central idea of this paper, we used the last 4 frames as an approximation to true-state. We are presently exploring other approaches to automatically learn good state representations in Atari Games.

We were able to investigate one of the interesting further-work ideas suggested by the reviewers. The use of empirical returns on games instead of UCT-values to train a regression based CNN. Empirical returns do not suffer from the bias of truncated trajectories in UCT but on the flip side they only provide values for the actions taken (while UCT computes values for all actions at the root node). Because we did not have to redo the slow UCT part, we could compare this new approach within the rebuttal week. Our results show that UCR (which uses UCT-values to train CNN via regression) outperforms this suggested method in all games but Seaquest. Additionally, UCC significantly outperforms this suggested method in all games (including Seaquest). We conjecture that with significantly larger training data sets the use of empirical return may indeed outperform UCR consistently. We will include these new results in a revision.

As noted by the reviewers there are many other interesting follow up questions; these are on our research path for current and future work. These include training of a single model across games, the use of features to improve performance (including as in Cobo etal’s work), theory and further empirical study on policy versus value function summarization into deep networks. Some of these questions are better answered on smaller domains; our focus in the submission was on success in the challenging (and increasingly benchmarked in RL) Atari games. We do conjecture that when there are only a small number of actions and long episodes/trajectories (as in Atari games) that the classification based policy learning in deep networks might work better than value-function learning in deep networks; though further work is needed to explore this question (e.g., perhaps both classification and regression can be used to learn even better features).

UCT is state of the art for Atari Games (when the emulator is allowed to be used as simulation model). We expect our results and methods to transfer to the use of other MCTS methods to provide training data for UCC.